# Dissipation-driven selection of states in non-equilibrium chemical networks

Daniel Maria Busiello [1✉], Shiling Liang [1], Francesco Piazza [2,3] & Paolo De Los Rios [1,4]

Life has most likely originated as a consequence of processes taking place in non-equilibrium conditions (e.g. in the proximity of deep-sea thermal vents) selecting states of matter that would have been otherwise unfavorable at equilibrium. Here we present a simple chemical network in which the selection of states is driven by the thermodynamic necessity of dissipating heat as rapidly as possible in the presence of a thermal gradient: states participating to faster reactions contribute the most to the dissipation rate, and are the most populated ones in non-equilibrium steady-state conditions. Building upon these results, we show that, as the complexity of the chemical network increases, the velocity of the reaction path leading to a given state determines its selection, giving rise to non-trivial localization phenomena in state space. A byproduct of our studies is that, in the presence of a temperature gradient, thermophoresis-like behavior inevitably appears depending on the transport properties of each individual state, thus hinting at a possible microscopic explanation of this intriguing yet still not fully understood phenomenon.

[1] Institute of Physics, École Polytechnique Fédérale de Lausanne - EPFL, Lausanne, Switzerland. [2] Centre de Biophysique Moléculaire (CBM), CNRS-UPR 4301, Rue C. Sadron, Orléans 45071, France. [3] Université d'Orléans, UFR CoST Sciences et Techniques, 1 rue de Chartres, Orléans 45100, France. [4] Institute of Bioengineering, École Polytechnique Fédérale de Lausanne - EPFL, Lausanne, Switzerland. ✉email: daniel.busiello@epfl.ch

The emergence of cellular life has likely been preceded by the appearance of molecular replicators, namely molecules able to use basic building blocks present in the environment to create copies of themselves[1]. RNA and other long macromolecules, such as proteins, are considered as the best candidates for the first replicators[2].

Although in the present oxidative conditions long biomolecules such as RNA are not thermodynamically stable, it is possible that they could be thermodynamically stable in the primordial Earth conditions (see refs. [3,4] for the still-open debate). Nonetheless, no conditions have been found to date such that their building blocks and their precursors could be thermodynamically stable and abundant enough to further proceed to their spontaneous polymerization[5] and subsequent self-replication. Relying on equilibrium thermodynamics is thus unlikely to provide a route to explain the emergence of life[6]. The potential relevance of non-equilibrium conditions in this context has also been highlighted in several recent works[7–9].

A different scenario is the possibility that, from the onset, external sources of energy might have driven prebiotic molecules away from equilibrium, allowing higher-energy states (i.e., more complex and/or longer molecules) to be abundant, even against their natural tendency to decay according to their equilibrium fate[10–12]. Consistently with these arguments, Braun and coworkers have for example shown that, in the presence of thermal gradients, the accumulation of molecules in regions of lower temperature (thermophoresis) increases polymerization beyond the prescriptions of mass-action kinetics at equilibrium[13].

In the present work, we hint at the possibility of broadening the perspective: external energy sources, here a thermal gradient, can tilt the populations of the different states that participate to a reaction network, by favoring states not only according to their energy, but also according to the dissipation along the pathways they are part of. In particular, we study linear reaction networks to highlight the basic rules deciding which states are the most favorable, relating them to kinetic and dissipation rates. Despite the simplicity of the model, it highlights the onset of unexpected features in non-isothermal chemistry, leading to potential applications in real-world examples[14].

## Results

**A temperature gradient favors states involved in faster reaction pathways.** The simple toy model that we propose here comprises three states, $A$, $B$, and $C$, which diffuse in space in the presence of a temperature gradient $\Delta T$. A pedagogical way to describe this system retaining all its essential non-equilibrium features is by means of a two-box model as depicted in Fig. 1a. Here diffusion is captured by allowing each state to move back and forth between the two boxes, with transport rates $d_A$, $d_B$, and $d_C$. The system evolves according to a Master Equation[15,16]:

$$
\begin{aligned}
\frac{dP(X_1)}{dt} &= \sum_{Y_1}\Big(k_{Y_1\to X_1}P(Y_1) - k_{X_1\to Y_1}P(X_1)\Big) + \\
&\quad + d_X(P(X_2) - P(X_1)) \\
\frac{dP(X_2)}{dt} &= \sum_{Y_2}\Big(k_{Y_2\to X_2}P(Y_2) - k_{X_2\to Y_2}P(X_2)\Big) + \\
&\quad + d_X(P(X_1) - P(X_2)),
\end{aligned}
\tag{1}
$$

where $X, Y = A, B, C$. To take into account the energy differences between the different states, the following relations between the transition rates must be respected[17–20]:

$$
\begin{aligned}
k_{A_1\to B_1} &= e^{(E_A-E_B)/k_B T_1}k_{B_1\to A_1} \\
k_{A_1\to C_1} &= e^{(E_A-E_C)/k_B T_1}k_{C_1\to A_1} \\
k_{A_2\to B_2} &= e^{(E_A-E_B)/k_B T_2}k_{B_2\to A_2} \\
k_{A_2\to C_2} &= e^{(E_A-E_C)/k_B T_2}k_{B_2\to A_2}.
\end{aligned}
\tag{2}
$$

with $T_1 = T_m + \Delta T/2$ and $T_2 = T_m - \Delta T/2$. We define the average temperature $T_m$ and the temperature gradient $\Delta T$, which is responsible for the maintenance of a non-equilibrium steady state. To further emphasize the effects that we want to highlight, we set the energies of the states $B$ and $C$ to be equal, $E_B = E_C$ (and $\Delta E = E_A - E_B = E_A - E_C$), with the additional condition on the height of the barrier that, *à la* Arrhenius, determines the velocity of the reactions

$$
k_{B_i\to A_i} = e^{-\Delta\epsilon/k_B T_i}k_{C_i\to A_i} \qquad \text{for } i = 1, 2
\tag{3}
$$

with $\Delta\epsilon = \Delta\epsilon_B - \Delta\epsilon_C > 0$. Equations (3) imply that, irrespective of the temperature (hence, in both boxes) the chemical transitions between $C$ and $A$ are faster than the ones between $B$ and $A$. Here, we are implicitly assuming that we can identify each state as a localized well in the chemical potential landscape, i.e., the activation energy for each reaction is much larger than the thermal energy available.

We are interested in the probability of finding the system in the lowest energy states, $B$ and $C$, at stationarity. In the following, $P(B)$ is identified as $P(B_1) + P(B_2)$, and analogously for $P(C)$. When equilibrium conditions are met, (namely $d_A = d_B = d_C = 0$ and/or $\Delta T = 0$), the system asymptotically converges to $P_i^{eq}(C) = P_i^{eq}(B) > P_i^{eq}(A)$ in each box and consequently $P^{eq}(B) = P^{eq}(C) > P^{eq}(A)$ overall. In non-equilibrium conditions the picture dramatically changes, because the energy symmetry between states $B$ and $C$ is kinetically broken. In order to emphasize the role of the barrier difference, $\Delta\epsilon$, we set all the transport rates to be equal, $d_A = d_B = d_C = d$. In this simple setting, away from equilibrium the state with the lowest energy barrier, $C$ in this case, is the most populated at steady-state in the presence of a temperature gradient. This is quantified by the ratio between the probabilities of the $C$ and $B$ states, $R_{CB} = P(C)/P(B)$, whose logarithm can be interpreted as the effective stabilization energy of $C$ relative to $B$ (Fig. 1b). $R_{CB}$ is always greater than 1, and in most physical cases reaches a maximum in the $d \to \infty$ limit, i.e., when diffusion between the two boxes is much faster than all other processes in the system. In this limit, it is possible to find the analytic expression of $R_{CB}$ for an arbitrary number $n$ of boxes:

$$
\lim_{d\to\infty} R_{CB} = \frac{\hat{k}_{B\to A}\hat{k}_{A\to C}}{\hat{k}_{C\to A}\hat{k}_{A\to B}}
\tag{4}
$$

with $\hat{k}_{X\to Y} = \sum_{i=1}^{n} k_{X_i\to Y_i}$.

The simple model that we have proposed here provides a clear example of kinetic symmetry-breaking due to the energy barriers, which is effective only in a non-equilibrium scenario[21]. In particular, the state which is more favorable away from equilibrium, $C$, participates in the reactions that, according to (3), are the fastest. The role of $\Delta\epsilon$ in the selection process is revealed in a small $\Delta T$ expansion of Eq. (4):

$$
R_{CB} = 1 + \frac{\Delta E \Delta T^2}{4T_m^4}\Delta\epsilon + \mathcal{O}(\Delta T^4).
\tag{5}
$$

As expected, the zeroth order is equal to 1, since at equilibrium the states $B$ and $C$ are equally populated. Furthermore, all the odd-order terms vanish because the selection of the fastest state cannot depend on the direction of the temperature gradient.

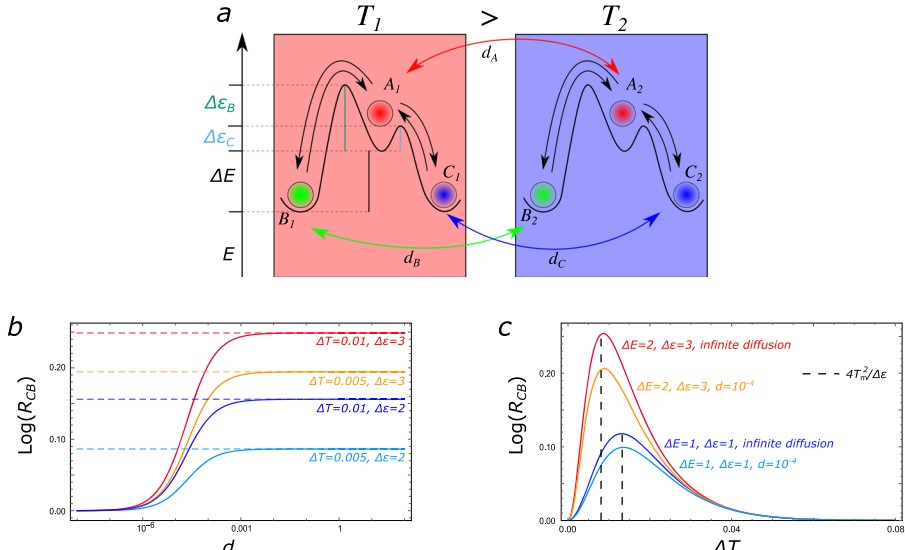

**Fig. 1 Selection of states in a three-state two-box model. a** A three-state chemical system diffusing in a temperature gradient, modeled as two connected boxes at different temperatures, $T_1 = T_m + \Delta T/2$ and $T_2 = T_m - \Delta T/2$. The states $B$ and $C$ have the same energy and the energy barrier between $A$ and $C$, $\Delta \epsilon_C$ (cyan/light gray), is lower than the one between $A$ and $B$, $\Delta \epsilon_B$ (green/dark gray), i.e., $\Delta \epsilon = \Delta \epsilon_B - \Delta \epsilon_C$. **b** The quantity $R_{CB} = (P(C_1) + P(C_2))/(P(B_1) + P(B_2))$ gauges the global non-equilibrium unbalance between $B$ and $C$. Under non-equilibrium conditions, $C$ is favored with respect to $B$, since it participates in faster reactions. Here, $T_m = 0.1$, $\Delta E = 2$, $\Delta \epsilon_C = 1$ and $d = d_A = d_B = d_C$, with $k_B$ set to 1. $R_{CB}$ is a monotonously increasing function of $d$, and the dashed lines indicate the limit for $d \to \infty$, computed in Eq. (4). **c** log ($R_{CB}$) is plotted as a function of the thermal gradient for different choice of parameters. The ones that remain fixed are as in **b**. Black dashed lines indicate the approximate value of the optimal gradient in both cases, Eq. (7). As we can see, it provides a good estimate of the actual value, even for finite diffusion.

**Optimal gradient for selection.** When $\Delta T \to 2 T_m$, the cold box tends to be kinetically inert. Hence, the system is dominated by the warm side, and all populations will eventually relax to their equilibrium value, corresponding to $R_{CB} = 1$. Since $R_{CB} = 1$ also when $\Delta T = 0$, and is always positive, it must have a maximum at a given $\Delta T^*$, suggesting that maximal selection would stem from a fine tuning of the parameters of the chemical network for any given $\Delta T$.

An approximate estimation of the optimal gradient for selection can be obtained in the infinite diffusion limit. We note that, up to the second order in $\Delta T$ in the exponents,

$$\hat{k}_{X \to A} \approx \frac{1}{2} e^{-\Delta \epsilon_X / T_m} \left( e^{\Delta \epsilon_X \Delta T / 2 T_m^2} + e^{-\Delta \epsilon_X \Delta T / 2 T_m^2} \right) \quad (6)$$

It is possible to identify two different regimes depending on whether $\theta_X \equiv \epsilon_X \Delta T / 2 T^2$ is much larger or much smaller than 1. Since $X = B, C$, we set as a control parameter the average between $\theta_B$ and $\theta_C$, thus estimating the optimal gradient as the crossing value for the two regimes:

$$\Delta T^* = \frac{4 T_m^2}{\Delta \epsilon_B + \Delta \epsilon_C}. \quad (7)$$

In Fig. 1c we show that indeed this formula provides a good estimate for the actual $\Delta T^*$.

**State selection is governed by dissipation.** An intuitive grasp of the mechanism leading to selection of the fastest state can be provided by Fig. 2a, where the direction of the currents have been highlighted. Thermal energy is converted into chemical energy, namely excess of $C$ over $B$, through diffusive cycles taking place in the system. Particles are heated up in the hot box ($B$ and $C$ toward $A$), thus absorbing heat, whereas they relax ($A$ to $B$ and $C$) in the cold box, thus releasing heat. This unbalance generates a current of $A$ from the warm to the cold box, where it splits preferentially along the faster decay path, that is, toward $C$, before being transported back into the hot box. Hence, $C$ is depopulated

during the heat absorption phase and populated when the heat is dissipated. This cycle is thus driven by the constant absorption and dissipation of energy, which is related to entropy production[16,22]:

$$\dot{S} = \sum_{i=1}^{2} \sum_{X=B,C} J_{A_i \to X_i} \ln \frac{k_{A_i \to X_i}}{k_{X_i \to A_i}}$$
$$= \Delta E \frac{\Delta T}{T_1 T_2} \left( J_{A_2 \to B_2} + J_{A_2 \to C_2} \right), \quad (8)$$

where $J_{A_i \to X_i} = k_{A_i \to X_i} P(A_i) - k_{X_i \to A_i} P(X_i)$ is the flux from $A_i$ to $X_i$, with $i$ indicating the box. We used $J_{A_1 \to X_1} = -J_{A_2 \to X_2}$ (Fig. 2), and the contributions from the interbox currents vanish because the rates in the two directions are equal. $\dot{S}$ is positive because the currents flow away from A at the colder temperature ($T_2$) and toward A at the warmer one ($T_1$).

Expanding Eq. (8) up to the second order in $\Delta T$, and using Eq. (5), we have:

$$R_{CB} \simeq 1 + \frac{\dot{S}}{\Delta E} \frac{1}{P^{eq}(A)} \frac{\Delta \epsilon}{e^{-\Delta \epsilon_B / T_m} + e^{-\Delta \epsilon_C / T_m}} \quad (9)$$

Despite the validity of this formula only for small gradients and fast diffusion, it suggests a correlation between $R_{CB}$, which quantifies selection, and $\dot{S}/\Delta E$, which is related to dissipation in the system. Intuitively, a similar relation could have been deduced noting that the probability fluxes towards $C$ are associated with the dissipation phase of the system.

In Fig. 2b, we show that indeed $R_{CB}$ and $\dot{S}/\Delta E$ are highly correlated for a set of (random) thermal gradients $k_B \Delta T = k_B(T_1 - T_2)$ and values of the typical energy scale $\Delta E$. Here it is clear that the gradient $\Delta T$ quantifies the available (thermal) energy driving the selection of the fastest state $C$ through dissipation. Indeed, as $\Delta T$ increases, the probability of escaping from $B$, diffusing, and populating $C$ by dissipating energy increases as well. However, as said before, when the

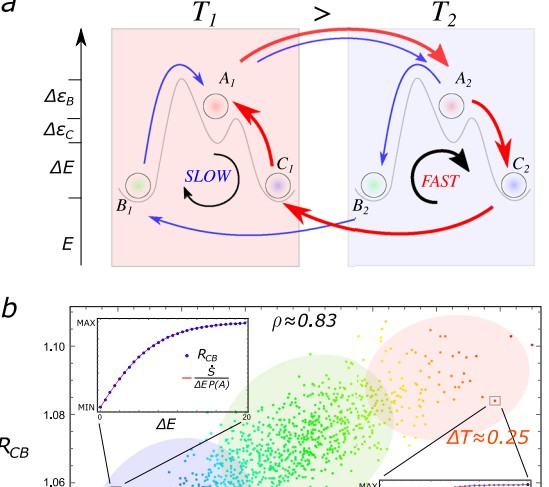

**Fig. 2 Features connecting dissipation and selection in a three-state two-box model. a** Diffusive cycles convert thermal energy into chemical energy. The direction and thickness of each arrow represent respectively direction and intensity of the net probability flux between two states. **b** Correlation between $R_{CB} = P(C)/P(B)$ and $\dot{S}/\Delta E$, which is the steady state entropy production divided by the characteristic energy scale of the system, for different values of $\Delta E$ and $\Delta T$. Here, $\Delta \epsilon = 2$, $T_m = 0.7$, $d \to +\infty$, both $\Delta E$ and $\Delta T$ have been drawn from a normal distribution of mean 1 and 0.2, with a variance of 0.1 and 0.02, respectively. Identical values of the gradient correspond to the same color. $\rho$ is the correlation coefficient. We report the approximate average $\Delta T$ among the values contained in each shaded area. *Insets* - Setting $\Delta T = 0.15$ (top) and $\Delta T = 0.25$ (bottom), we show that $R_{CB}$ and $\frac{\dot{S}}{\Delta E P(A)}$ exhibit the same behavior as a function of $\Delta E$, when plotted within the same range, extending Eq. (9) beyond the small gradient regime. The Boltzmann coefficient has been taken equal to 1 for simplicity.

gradient is too high, chemical selection is abolished. Remarkably, fixing the thermal gradient, $R_{CB}$ is always strongly correlated with the steady-state entropy production, rescaled by the probability of the high-energy state in analogy with Eq. (9), as a function of the energy $\Delta E$ (Insets of Fig. 2b). In the Supplementary Discussion, we show that the correlation is preserved also in the case of finite diffusion and for for different values of energy barriers and temperature gradient (see Figs. S2 and S3).

**Characteristic lengthscale for selection.** Extending this two-box model to a thermal gradient in continuous space is of course more realistic, and reveals further features that are inaccessible to the discrete box description. In continuous space (say, $x \in [0, 1]$), the system evolves according to the differential Chapman–Kolmogorov equation[15]:

$$\partial_t p_X(x) = \sum_Y \left( k_{Y \to X}(x) p_Y(x) - k_{Y \to X}(x) p_X(x) \right) + D_X \partial_x^2 p_X(x),$$

(10)

where $X, Y = A, B, C$. We impose no-flux boundary conditions, i.e., $\partial_x p_X(0) = \partial_x p_X(1) = 0$. In Eq. (10), the Laplacian captures diffusion while the part involving discrete transitions captures the chemical reactions between species, which are governed by rates

analogous to the ones introduced before:

$$k_{A \to B}(x) = e^{(E_A - E_B)/k_B T(x)} k_{B \to A}(x)$$
$$k_{A \to C}(x) = e^{(E_A - E_C)/k_B T(x)} k_{C \to A}(x),$$

(11)

with the additional condition on the energy barriers:

$$k_{B \to A}(x) = e^{-\Delta \epsilon/k_B T(x)} k_{C \to A}(x).$$

(12)

Also in this case, the transport coefficient is the same for all states: $D_X \equiv D$, $\forall X$. In what follows $P(X) = \int dx p_X(x)$ (note that we use $p$ for the space dependent distribution, and $P$ to indicate their integral over space).

Although it is difficult to solve Eq. (10) analytically for any value of the parameters, approximate solutions can be worked out in selected cases. The limit of large diffusion ($D \to \infty$), which is analogous to the case of infinitely fast transport between the two boxes analyzed above, can be tackled using the standard approach of time-scale separation[15,23]. To the leading orders in $1/D$, the solution is uniform in space, and $R_{CB}$ is the same as in Eq. (4), with $\hat{k}_{X \to Y} = \int dx\, k_{X \to Y}(x)$.

The case of a linear temperature gradient $T(x) = T_0 + \Delta T \cdot x$ can also be analytically explored for small $\Delta T$. Expanding all rates and probabilities in powers of $\Delta T$ as

$$k_{X \to Y} = \sum_n \frac{1}{n!} x^n \Delta T^n \partial_T^n k_{X \to Y}|_{\Delta T = 0}$$
$$p_X(x) = \sum_n \Delta T^n p_X^{(n)}(x),$$

(13)

inserting them in (10) and solving it order by order it is easy to obtain at 0th order

$$p_B^{(0)} = p_C^{(0)} = \frac{e^{\Delta E/k_B T_0}}{2 e^{\Delta E/k_B T_0} + 1},$$

(14)

which is the equilibrium solution for $\Delta T = 0$.

Up to second order, $R_{CB}$ is

$$R_{CB} = 1 + \frac{\Delta T^2}{2 P_B^{(0)}} \left( P_C^{(2)} - P_B^{(2)} \right),$$

(15)

where $P_X^{(n)}$ is defined as the integral of $p_X^{(n)}(x)$ over the whole domain. After a further expansion in $\Delta \epsilon$, i.e., the symmetry between $B$ and $C$ is only infinitesimally broken by the kinetics, we obtain

$$P_C^{(2)} - P_B^{(2)} = \frac{\Delta E}{T_0^4} P^{eq} L_s^2 \left( 1 - 2 L_s \tanh\left( \frac{1}{2 L_s} \right) \right) \Delta \epsilon,$$

(16)

with $L_s = \sqrt{D/(k_{B \to A} + 2 k_{A \to B})}$ and $P^{eq} = P_B^{(0)} = P_C^{(0)}$. This difference is always positive, implying that states participating in fast reactions are always favored in this system, and it vanishes when $D \to 0$, as expected because the system locally relaxes at equilibrium. In particular, $L_s$ represents a typical length-scale that can be interpreted as the space traveled by the system between two state transitions, namely the distance below which the system can absorb and dissipate energy, thus setting a lengthscale for dissipation-driven selection.

Analogously to the two-box scenario, an optimal gradient can be appreciated also in this case. Since the system lives in a continuous domain, its physical origin is slightly different. When $\Delta T \to \infty$, all states tends to be equally populated, i.e., $R_{CB} \to 1$, abolishing chemical selection. The positivity of $R_{CB}$, and the fact that it goes to zero when for a vanishing gradient, leads to the existence of a maximum at $\Delta T^*$. In the Supplementary Methods, we present a more in-depth discussion on this (see Fig. S1).

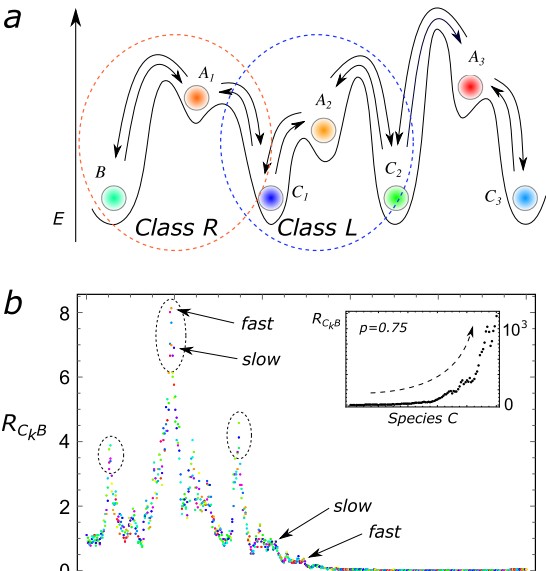

**Fig. 3 Selection of states in a chain of three-state subsystems in a thermal gradient. a** Chain of three concatenated three-state chemical networks, each similar to the one in Fig. 1a. The orange circle indicates a subsystem belonging to the class $R$, with the fast transition on the right branch, while the blue circle indicates a subsystem whose fast transition is on the left branch (class $L$). **b** $R_{C_k B} = P(C_k)/P(B)$ as a function of the states $C_k$. The selection of states does not depend only on their transitions being fast or slow with respect to the neighboring reactions, but on all rates of the network. Here $k_B = 1$, $T = 0.7$, $\Delta T = 0.2$, $\Delta E_k \sim U([1, 10])$, $\Delta \epsilon_{slow}^{(i)} \sim U([3, 6])$ and $\Delta \epsilon_{fast}^{(i)} \sim U([2, 3])$, where $U$ is the uniform distribution. Each subsystem belongs to class $L$ with probability $p = 0.5$, and to class $R$ with probability $1 - p$. Inset—$R_{C_k B}$ as a function of the species $C_k$ for the same parameters as in the main panel, but $p = 0.75$. The predominance of subsystems belonging to the class $R$ leads to a directional exponential growth.

**Non-trivial selection for more complex reaction-network topologies**. How do the features of a simple three-state system extend to more complex network topologies? Here we study a chain of connected chemical reactions in an energy landscape, looking at the propagation of the local selection process along the chain, eventually leading to runaway and/or localization phenomena in the population of states.

We focus to the two-box scenario, which, as shown above, recapitulates most of the dissipation-driven selection phenomenon while being easier to analyze, in the limit of infinitely fast transport between the boxes. We consider a reaction network as the one sketched in Fig. 3a, which can also diffuse between two boxes at different temperatures as in Fig. 1. We can distinguish two different classes of three-state subsystems, with the faster reaction either on the right branch (henceforth indicated as $R$, encircled by an orange dashed line in Fig. 3a), analogously to the three-state system depicted in Fig. 1a, or on the left ($L$, encircled by a blue dashed line in Fig. 3a). All lower energy states have the same energy, while the high energy state in each subsystem is characterized by a different energy $\Delta E_i$, and two different barriers $\Delta \epsilon_{slow}^{(i)}$ and $\Delta \epsilon_{fast}^{(i)}$, with $\Delta \epsilon_i = \Delta \epsilon_{slow}^{(i)} - \Delta \epsilon_{fast}^{(i)}$, mimicking the presence of a non-trivial underlying energy landscape.

We have already computed $R_{CB}$ in Eq. (4), in the limit of infinitely fast diffusion. It quantifies the ratio between the population of two adjacent states, the fast over the slow one. It is possible to see from the Master Equation for the whole system

in Fig. 3a, that the same relation holds between any two adjacent states in each subsystem. Since, we want to compute the population of each single species along the chain, we use $B$ as our reference state. The ratio between $P_{C_k}$ and $P_B$ is:

$$R_{C_k B} = R_{C_1 B} \prod_{l=2}^{k} R_{C_l C_{l-1}} \qquad (17)$$

If there are $n_k^{(L)}$ subsystems belonging to the class $L$, and $n_k^{(R)} = k - n_k^{(L)}$ subsystems to the class $R$, then Eq. (17) becomes:

$$R_{C_k B} = \prod_{i=1}^{n^{(R)}} R_i^{(R)}(\Delta E_i, \Delta \epsilon_{slow}^{(i)}, \Delta \epsilon_{fast}^{(i)}) \prod_{i=1}^{n^{(L)}} R_i^{(L)}(\Delta E_i, \Delta \epsilon_{slow}^{(i)}, \Delta \epsilon_{fast}^{(i)}) \qquad (18)$$

with both $R_i^{(R)}$ and $R_i^{(L)}$ given by Eq. (4).

To simulate a generic chain of chemical reactions, we assign each subsystem to class $L$ with probability $p$, and to class $R$ with probability $q = 1 - p$. We then draw $\Delta E_i$, $\Delta \epsilon_{slow}^{(i)}$ and $\Delta \epsilon_{fast}^{(i)}$ from three different distributions, $P(\Delta E)$ and $P(\Delta \epsilon_{slow})$ and $P(\Delta \epsilon_{fast})$, respectively (details in the caption of Fig. 3). As we can see from Fig. 3b, even in the simple case in which $p = q = 1/2$, and both distributions are uniform, a localization phenomenon in the population of the states can spontaneously arise, where the favorability of an individual state does not depend only on its fast/slow status with respect to the adjacent reactions, but depends instead on the full path of reactions connecting it to the reference state, and hence on the full energy landscape. If all the fast reactions are on the same side of each three-state subsystem (all reactions of type $R$ or of type $L$), the population of states $C_k$ can become exponentially different from the one of $B$, as highlighted in the inset of Fig. 3b.

Also in this case, the selection for the most probable states is determined by dissipation. The argument outlined for the simple three-state system can be easily generalized in the case of infinitely fast transport between the boxes: $R_{C_k B}$ simply corresponds to the product of all the transition rates directed from $B$ to $C_k$ belonging to the path connecting the two, divided by the same product in the opposite direction. As a consequence, the states that will eventually be the most populated ones (with respect to a reference state $B$) are those whose connecting path to $B$ have the fastest dissipation. In the Note 1, we study a tree-like topology, showing that the kinetic properties of the path connecting lower and higher energy states become relevant to determine steady selection under non iso-thermal conditions (see Fig. S4). However, when the topology is further complicated, several distinct paths can connect the same pair of states, and all the transition rates will eventually contribute to determine a ranking for steady-state populations. In this case the determination of the fastest dissipating states becomes difficult, and we leave for future works the development of an efficient technique to tackle this problem.

**Selection under time-periodic variations of temperature**. Another common method to maintain a system out-of-equilibrium is to periodically vary some external parameters, allowing it to reach a time-periodic non-equilibrium state[24,25]. We imagine to have the three-state system detailed above, coupled to a reservoir whose temperature is varied periodically in time, with a period $\tau$:

$$T(t) = T_m + \eta_\tau(t)\Delta T, \qquad (19)$$

with $\overline{\eta_\tau(t)} = 0$ and $\overline{\eta_\tau(t)\eta_\tau(t')} = \delta(t - t')$, where the overline indicates the temporal average over one period. In analogy with

Eqs. (11) and (12), the time-dependent transition rates satisfy:

$$k_{A \to B}(t) = e^{(E_A - E_B)/k_B T(t)} k_{B \to A}(t)$$
$$k_{A \to C}(t) = e^{(E_A - E_C)/k_B T(t)} k_{C \to A}(t)$$
$$k_{B \to A}(t) = e^{-\Delta\epsilon/k_B T(t)} k_{C \to A}(t). \tag{20}$$

The inverse period $\tau^{-1}$ plays the same role as the diffusion rate $d$ in the system in Fig. 1a. It favors cycles of constant absorption and dissipation of energy, hence igniting a chemical selection among species. The latter is defined as the ratio between temporal averaged probabilities. Hence, in the limit of infinitely fast driving, we have an equation similar to Eq. (4).

$$\lim_{\tau \to 0} R_{CB} = \frac{\overline{P(C)}}{\overline{P(B)}} = \frac{\overline{k_{B \to A}}}{\overline{k_{C \to A}}} \frac{\overline{k_{A \to C}}}{\overline{k_{A \to B}}} \tag{21}$$

For small values of the gradient $\Delta T$, by means of a perturbative analysis, we solve the system order by order, as for the case of a continuous gradient. The zeroth-order in $\Delta T$ corresponds to the equilibrium solution. Remarkably, even in this case the first-order correction vanishes and, up to the second order in $\Delta T$, we have:

$$R_{CB} = 1 + \frac{\Delta E \Delta T^2}{T_m^4} \Delta\epsilon \tag{22}$$

Its expression in terms of time-averaged entropy production is

$$R_{CB} = 1 + \frac{\overline{\dot{S}(t)}}{\Delta E} \frac{1}{P^{eq}(A)} \frac{\Delta\epsilon}{e^{-\Delta\epsilon_B/T_m} + e^{-\Delta\epsilon_C/T}} \tag{23}$$

Equations (22) and (23) are analogous to Eqs. (5) and (9), and notably they are valid in the fast driving regime. We highlight the analogy between temporal cycles and diffusive cycles of the previous setting: they both force the system to explore different temperatures. In the Supplementary Methods we present the detailed derivation of these results.

**Emergence of thermophoresis-like behavior.** So far, we have assumed that all the species move between the boxes (or diffuse in space) at the same rate, and as a consequence the probability to be in each box, summed over the different states, is always equal to 1/2 (or uniform in continuous space). Although relaxing this hypothesis does not significantly change the overall picture of dissipation-driven selection, a novel phenomenon appears, that we are compelled to report for its potential implications: we find that, even in the simple two-box scenario, there is an accumulation of the population in one of the boxes. The description of this effect is surprisingly similar to thermophoresis, which refers to the accumulation of molecules on either the cold or warm side in presence of a thermal gradient. Mathematically, at stationarity, thermophoresis is usually described through a diffusive equation[26],[27]:

$$\nabla c = -S_T c \nabla T, \tag{24}$$

where $c$ is the concentration of particles, and $S_T$ is the so-called Soret coefficient, which can be positive or negative. Even if extensively described through effective equations, a microscopic understanding of this behavior is still lacking[28]–[30]. The present approach might serve as a complementary perspective for this intriguing phenomenon.

To fix the ideas, consider the discrete-state system sketched in Fig. 1a. We consider the ratios $d_B/d_A$ and $d_C/d_A$ as measures of the unbalance of transport properties of different species. The probability of being in box $i$ is $P_i = P(A_i) + P(B_i) + P(C_i)$. In a discrete box scenario, Eq. (24) can be rewritten as:

$$\Delta P = P_2 - P_1 = -S_T \left(\frac{P_1 + P_2}{2}\right) \Delta T \tag{25}$$

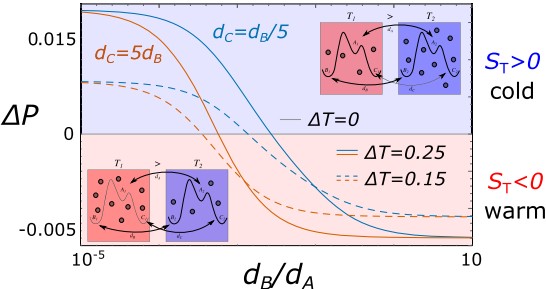

**Fig. 4 Thermophoresis as selection in space for a three-state two-box model.** Difference between the probability of being in each box, $\Delta P = P_2 - P_1$, as a function of $d_B/d_A$ (in log-scale), for $d_C = d_B/5$ (blue curves) and $d_C = 5 d_B$ (vermilion curves). Different values of the gradient $\Delta T$ are shown. When $\Delta P$ is positive, the particles (independently of the species) accumulate on the cold side (blue/right box), and the Soret coefficient $S_T$ is positive. On the contrary, the particles are more abundant in the warm side (red/left box) for negative $\Delta P$, corresponding to negative values of $S_T$. In this example, we set $T_m = 0.7$, $\Delta E = 0.1$, $\Delta\epsilon = 2$ and, $d_A = 0.01$. $k_B$ has been taken equal to 1 for simplicity.

Since for infintely fast transport the system will end up equally populating both boxes, we need to consider finite transport. $\Delta P = P_2 - P_1$ is represented in Fig. 4 as a function of $d_B/d_A$, for two different choices of $d_C/d_A$, and for different values of $\Delta T$. Clearly, in the absence of a thermal gradient there is no thermophoresis, while a difference between $T_1$ and $T_2$ induces an accumulation of particles on the warm or cold side. When the transport coefficients are small compared to all the other transition rates in the system, the Soret coefficient can be estimated to be equal to:

$$S_T = \frac{\left(2 - \frac{d_B + d_C}{d_A}\right) \Delta E \, e^{\Delta E/k_B T_m}}{\left(1 + 2e^{\Delta E/k_B T_m}\right) \left(\frac{d_B + d_C}{d_A} e^{\Delta E/k_B T_2} + 1\right) k_B T_m^2}. \tag{26}$$

As can be seen from (26), the sign of $S_T$ depends on the values of the transport coefficients of the different states, and it thus inextricably links transport to the internal kinetics in chemical space. Indeed, even a simple two-state system exhibits thermophoresis, as long as the two states have different transport coefficients (see Supplementary Note 2 and Fig. S5).

In line with the leit-motif of this work, we highlight here that thermophoresis can again be seen as a selection process in position, rather than in state, space. It is driven by the dissipation of thermal energy, and the kinetic symmetry-breaking is induced by the asymmetry of transport rates.

## Discussion

Non-equilibrium conditions can trigger stabilization effects in molecular systems[31]–[34]. In a similar fashion, here we have shown that high energy states can be stabilized out-of-equilibrium, by continuously dissipating energy supplied from an external source, a temperature gradient in our case. In particular, the deviation with respect to equilibrium directly correlates with dissipation, which is kinetically controlled by the rates of the system. Hence, the core ingredient is the breakdown of kinetic symmetry in the reaction rates: while at equilibrium the energies are the only relevant quantities, away from equilibrium the kinetics plays a fundamental role. Here, we have proposed simple reaction networks that could be investigated to reveal how selection and dissipation are intimately related. Furthermore, because of their simplicity, these models can be analytically and numerically solved and, importantly, are amenable of experimental validation. As a byproduct of our study, we have presented a thermophoresis-like behavior emerging as a spatial selection

process. This is induced, again, by kinetic symmetry-breaking, in this case in the diffusion coefficients of different states. It is also worth noting that the relation between selection and dissipation stems from the thermodynamic necessity to transport heat from the warm to the cold side of the system. In this respect, selection becomes a necessary consequence of thermodynamics.

From a broader perspective, this work could provide a novel framework to develop schemes aimed at explaining the sustained abundance of otherwise only metastable molecules, which are necessary intermediates for the spontaneous synthesis of more complex macromolecules that, in turn, could lead to the first replicators. In this respect, the approach here presented recently stimulated a possible solution to the furanose conundrum[14]. Hence, we are convinced that our results could represent an important previously unreported ingredient to connect the origin of life problem to the physical questions of what is possible in non-equilibrium conditions[35,36], and what are the basic microscopic (molecular) rules governing the emergent phenomena.

## Data availability
No datasets were generated or analysed during the current study.

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

## Acknowledgements
We acknowledge A. Maritan, F. Stellacci, C. Jarzynski, and D. Astumian for useful discussions, M. A. Younan for the help in developing the expansion for small value of the gradient in the continuous-space description and for insightful observations, V. Ouazan for thoughful comments. The authors thank the Swiss National Science Foundation for support under grant 200020_178763.

## Author contributions
D.M.B. and S.L. performed numerical analysis, all authors carried on analytical calculations. P.D.L.R. conceived and designed the work, with the help of F.P. and D.M.B. All authors wrote the manuscript.

## Competing interests
The authors declare no competing interests.
