## [Peer Review File · Communications Chemistry]

This manuscript has been previously reviewed at another Nature Research journal. This document only contains reviewer comments and rebuttal letters for versions considered at Communications Physics.

Reviewers' comments:

Reviewer #1 (Remarks to the Author)

The authors did not formulate any significant objection to the different points of my severe criticism. Most of the answers are of the type "the referee is right but...". I maintain my position. This paper is an exploration of the stochastic thermodynamics of simple linear networks in non-isothermal regimes. I would consider such a paper for a Phys. Rev. E for instance, but under no circumstance Nature Communications. The paper remains misleading despite limited efforts to tone down the most unsubstantiated claims. Linear chemical networks have little to know relevance for biochemistry or evolution. Even within that very simple class of networks the results presented remain of limited scope.

The fact that Reviewer 2 asks for proof and not "just statements and simulation" is very consistent with my criticism.

Reviewer 3 asks if "the model (can) be translated to cases that can be found in nature?". The answer is no for any meaningful biochemical problem. He/she asks if any connection with Turing patterns can be made. The answer is a clear no. Nonlinearities and space dependence are essential for such phenomena to arise. None of this can happen in linear networks which simply exponentially relax to unique stationary states.

Reviewer #2 (Remarks to the Author)

I think the authors addressed my previous concerns. ALthough I still am not sure about the significance of the work, I can suggest its publication and let the time decide.

Reviewer #3 (Remarks to the Author)

The paper has now improved with respect to the first version. I still think the authors should improve Fig.2A by including in the paper excerpts of the authors response to my comment. Otherwise the figure remains confusing.

Detailed answer to Referee #1

The authors did not formulate any significant objection to the different points of my severe criticism. Most of the answers are of the type "the referee is right but...". I maintain my position.

We would like to make it clear that most of the answers could not have been different from "the referee is right, but...", since most of the points were ideological claims, rather than constructive and technical criticisms. With respect to the latter, we provide a careful explanation, and substantially modified the manuscript after the first round of reviews.

This paper is an exploration of the stochastic thermodynamics of simple linear networks in non-isothermal regimes. I would consider such a paper for a Phys. Rev. E for instance, but under no circumstance Nature Communications. The paper remains misleading despite limited efforts to tone down the most unsubstantiated claims. Linear chemical networks have little to know relevance for biochemistry or evolution. Even within that very simple class of networks the results presented remain of limited scope.

We do not understand how unsubstantiated claims, if any, may make misleading the paper. We should deduce that the main source of criticism stems from the fact that the referee #1 does not consider the presented result of any importance in the context of the origin of life and prebiotic chemistry.

- i) *We decided to maintain all references to the origin of life problem throughout the manuscript, since this topic represented the first stimulus for this work. We are also convinced that theoretical advances in non-isothermal chemistry can be relevant for this field, and we want to highlight this aspect in the manuscript.*
- ii) *We are aware of the fact that multi-molecular reactions can give rise to higher levels of complexity but, precisely because of their complexity, the roles of basic processes leading to selection of states are more difficult to disentangle.*
- iii) *We tone down some sentences in the revised manuscript, highlighting that the work has to be considered a first step in the direction of understanding the emergence of complexity from non-equilibrium conditions. We also evidenced that our finding, despite its simplicity, is a previously unreported ingredient to connect the origin of life problem to physical questions.*
- iv) *We added a recent manuscript of ours (Ref. [23]) in which D-ribose isomerization is studied, both theoretically and experimentally, using the framework we introduced. This system plays a key role in prebiotic chemistry, and as the referee can imagine, it is not the unique example.*

The fact that Reviewer 2 asks for proof and not "just statements and simulation" is very consistent with my criticism.

There is not much to say, apart that Referee #2 suggests publication in this second round. Moreover, it is not entirely clear to what kind of "proof" Referee #1 is referring to, and for what exactly. Additionally, we should add that our paper does not report "statements", but a thorough and rigorous mathematical analysis of a well-defined physical model, upon which our predictions are based.

Reviewer 3 asks if "the model (can) be translated to cases that can be found in nature?". The answer is no for any meaningful biochemical problem. He/she asks if any connection with Turing patterns can be made. The answer is a clear no. Nonlinearities and space dependence are essential for such

phenomena to arise. None of this can happen in linear networks which simply exponentially relax to unique stationary states.

We express our deepest concern about this comment at several levels.

First and foremost, we did not answer “No” to this comment of the Referee #3 in the first round of review. On the contrary, we declared that we were actively working (theoretically and experimentally) on problems relevant to prebiotic chemistry based on linear reaction networks, upon being contacted by two experimental groups having manifested their interest in our manuscript on arXiv (Ref. [23]). Hence, we are unsure whether an ideological bias or sheer superficiality originated this brusque, patently false statement.

Second, if instead “No” is not ours, but the referee’s rhetoric answer, we challenge them to prove that no meaningful biochemistry problem could be affected by our results in any relevant way.

Finally, we believe that the impact of our work mainly resides in the possibility to be an eye-opening perspective especially for those researchers (as the referee) convinced that the intricate world of prebiotic chemistry, and its emergent complexity, cannot be understood starting from simple thermodynamic models.

REVIEWERS' COMMENTS:

Reviewer #3 (Comments to Authors):

Editorial Note: This reviewer has not provided any further comments to the authors.